# Fibrosis Is a Basement Membrane-Related Disease in the Cornea: Injury and Defective Regeneration of Basement Membranes May Underlie Fibrosis in Other Organs

**DOI:** 10.3390/cells11020309

**Published:** 2022-01-17

**Authors:** Steven E. Wilson

**Affiliations:** Cole Eye Institute, I-32, Cleveland Clinic, 9500 Euclid Ave, Cleveland, OH 44195, USA; wilsons4@ccf.org

**Keywords:** fibrosis, basement membranes, myofibroblasts, fibrocytes, corneal fibroblasts, TGF beta, collagen type IV, cornea, skin, lung, liver

## Abstract

Every organ develops fibrosis that compromises functions in response to infections, injuries, or diseases. The cornea is a relatively simple, avascular organ that offers an exceptional model to better understand the pathophysiology of the fibrosis response. Injury and defective regeneration of the epithelial basement membrane (EBM) or the endothelial Descemet’s basement membrane (DBM) triggers the development of myofibroblasts from resident corneal fibroblasts and bone marrow-derived blood borne fibrocytes due to the increased entry of TGF beta-1/-2 into the stroma from the epithelium and tears or residual corneal endothelium and aqueous humor. The myofibroblasts, and disordered extracellular matrix these cells produce, persist until the source of injury is removed, the EBM and/or DBM are regenerated, or replaced surgically, resulting in decreased stromal TGF beta requisite for myofibroblast survival. A similar BM injury-related pathophysiology can underly the development of fibrosis in other organs such as skin and lung. The normal liver does not contain traditional BMs but develops sinusoidal endothelial BMs in many fibrotic diseases and models. However, normal hepatic stellate cells produce collagen type IV and perlecan that can modulate TGF beta localization and cognate receptor binding in the space of Dissé. BM-related fibrosis is deserving of more investigation in all organs.

## 1. Introduction

Fibrosis is a common terminal pathology for numerous insults in most, if not all, organs. For example, in the lung, fibrosis can be triggered by radiation, infection, toxic exposures, hypersensitivity pneumonitis or unknown factors underlying idiopathic pulmonary fibrosis [1]. Similarly, many insults can produce scarring stromal fibrosis in the cornea, including viral infections, bacterial infections, trauma, chemical burns, and surgical procedures [2]. While fibrosis often represents an end stage of irreversible organ dysfunction, there are specific conditions where fibrosis may reverse spontaneously if the inciting factors are eliminated. For example, myofibroblast-related scarring stromal fibrosis of the cornea after laser vision correction photorefractive keratectomy (PRK) often resolves spontaneously over time measured in months to years in both humans and rabbits [3,4].

Major advances have been made in understanding the factors underlying corneal stromal fibrosis over the past few years. This research pointed to the critical roles of the two major basement membranes (BMs) of the cornea, the epithelial basement membrane (EBM) underlying the epithelium [5] and Descemet’s basement membrane (DBM) overlying the corneal endothelium [6,7], in the pathophysiology of stromal fibrosis. Both the corneal EBM and DBM modulate the passage of transforming growth factor (TGF) beta-1 and TGF beta-2 into the central stroma from the tears and epithelium [5] or residual corneal endothelial cells and the aqueous humor [7], respectively. In both locations, injury and defective or delayed regeneration of the corneal BMs leads to penetration of high levels of the TGF beta into the stroma that drive the development of myofibroblasts from at least two different precursor cells. The pathophysiological mechanisms leading to fibrosis in the cornea are likely relevant to fibrosis that develops in many other organs. This review will detail the pathophysiology of fibrosis in the cornea and then provide examples where BM injury and defective regeneration can underlie the development of fibrosis in other organs.

## 2. BM-Related Corneal Fibrosis

The cornea is an exceptional model to study fibrosis due to its relative simplicity, normal transparency, available imaging modalities, and large variety of reproducible wounding models. Corneal fibrosis responses can be separated into anterior EBM-related and posterior DBM-related fibrosis, although severe injuries and infections often involve both BMs and some fibrosis producing infections or immune-related diseases can enter from the peripheral limbal blood vessels.

### 2.1. Normal Cornea Structure and Transparency

The normally transparent cornea (Figure 1A) is composed primarily of three tissues, the epithelium, stroma, and corneal endothelium (Figure 2). The approximately 50 µm thick corneal epithelium is a 5 to 7 layer thick nonkeratinized, stratified, squamous epithelium that lies atop the EBM (Figure 2) and is bathed in tears produced by the accessory and main lacrimal glands. The stroma is approximately 300 to more than 600 µm thick, depending on the species, and is populated by keratocan-positive keratocyte fibroblastic cells that function to maintain the unique packing of uniform diameter stromal fibers (Figure 3A,B) that provides the cornea its transparency. The stromal ECM is primarily composed of collagen type I, along with smaller amounts of collagen type III, IV, V, VI, VIII and XII [8]. The extracellular matrix (ECM) between the corneal stromal fibrils, that is sometimes referred to as the “ground substance”, contains small leucine-rich proteoglycans (SLRPs), including decorin, biglycan, lumican, keratocan, and fibromodulin [8]. Some species, including humans, have an acellular condensation of the anterior stroma termed Bowman’s layer [9]. The corneal endothelium [10], unlike vascular endothelium, develops from neural crest and is a monolayer of cells that lies posterior to DBM (Figure 2) [11]. The corneal endothelium cooperates with keratocytes to produce the DBM during development and after injury [7]. The proliferative capacity of the corneal endothelial cells may vary between species, with human endothelial cells thought to have a relatively low capacity to proliferate, but this may be related to the ages of the animals studied [10].

The normal central cornea is avascular, but bone marrow-derived cells, including fibrocytes, migrate into the stroma from the edge of the cornea (limbus) after corneal injuries [12]. The corneal stroma is also richly innervated, primarily with sensory nerves that arborize and terminate in the basal epithelium [13]. There are also normally small numbers of immune cells, including resident macrophages and Langerhans cells in the cornea [14].

### 2.2. The Corneal Wound Healing Response to Injury

The first observable stromal cellular change after corneal injury is apoptosis (Figure 4) of the keratocytes in proximity to the injury to the epithelium [15] or the endothelium [16] that is mediated by interleukin (IL)-1 alpha released by the injured epithelial and/or endothelial cells and the activated Fas/Fas ligand system [17,18]. The extent of the apoptosis response is proportional to the severity of the injury to the epithelium and/or endothelium [19], and this response is thought to have evolved as an innate response to produce a cellular firebreak in rejoinder to viral infections to the epithelium and/or endothelium that have the capacity to spread to stromal cells and into the eye [20].

Injury to the epithelium and underlying EBM results in the entry of large amounts of TGF beta-1 and TGF beta-2 (Figure 5) from the corneal epithelium and tears into the stroma [5], in addition to other growth factors such as the platelet-derived growth factor (PDGF). Similarly, injury to the corneal endothelium and overlying DBM results in the entry of large amounts of TGF beta-1 and TGF beta-2 into the stroma from the aqueous humor in the anterior chamber of the eye and residual peripheral corneal endothelial cells [7]. In both the epithelial-stromal and endothelial-stromal injuries some stromal cells also begin to produce TGF beta isoforms, but this production is relatively limited compared with the TGF beta that enters from tears, epithelium, endothelium and aqueous humor [5,7]. The TGF beta-1 and -2, along with PDGF, trigger keratocytes in proximity to the injury, that escape the initial wave of keratocyte apoptosis, to differentiate into vimentin-positive, keratocan-negative corneal fibroblasts. These corneal fibroblasts, along with fibrocytes that enter the stroma from the limbal blood vessels [12,21], begin a developmental program to transition into alpha-smooth muscle actin (SMA)-positive, desmin-positive, vimentin-positive, keratocan-negative myofibroblasts, and that development only continues as long as requisite levels of TGF beta are available in the stroma where these precursors exist. Otherwise, if stromal TGF beta levels decline, the precursors undergo apoptosis or transition back to their cell types of origin [5,7]. That myofibroblast development from precursor corneal fibroblasts and fibrocytes occurs over a period of weeks to months depending on the severity of injury, the localized concentration of TGF beta-1 and TGF beta-2 in the stroma [5,7], and the species. For example, after high correction photorefractive keratectomy (PRK) surgery, scarring stromal fibrosis develops 3 to 4 weeks after surgery in rabbits [5] but typically does not develop until three to 6 months after surgery in humans [3]. Critically, whether the development of corneal fibroblast- and fibrocyte-derived precursor cells receive sufficient and ongoing levels of TGF beta-1 and TGF beta-2 depends on whether, or not, the EBM and/or DBM regenerate their normal structures and functions in a timely manner (or in the case of DBM is replaced surgically by transplantation) [5,7]. In turn, whether the EBM regenerates in a timely manner depends on the severity of the injury (and, therefore, the level of the initial keratocyte apoptosis response), the irregularity of the stromal surface induced by the trauma or surgery (that interferes mechanically with EBM regeneration), and likely genetic factors [22].

### 2.3. Minor Injuries and Non-Fibrotic Healing in the Anterior Cornea

Relatively minor injuries to the anterior cornea, such as corneal abrasions or laser vision correction PRK for low myopia, usually heal with little stromal opacity (Figure 1B) and no stromal fibrosis (Figure 6B) [5]. The priority to prevent fibrosis after these injuries is for the epithelium to first close within a period of 1 to 2 weeks [5], and therefore, that a persistent epithelial defect does not develop [23]. This is because the epithelial cells, and not keratocytes or corneal fibroblasts, at least early in the regeneration process, produce self-polymerizing laminins 511 and 521 that initiate BM regeneration [24], and trigger the subsequent binding of other BM components, such as perlecan and nidogens, to form the nascent EBM [5]. Thus, no EBM regeneration occurs in an area of the cornea where the epithelium does not close, and if it does not close, stromal fibrosis invariably develops in that area [23]. The critical importance of the fully regenerated EBM (and DBM that will be discussed later in this paper) is that it contains the components perlecan and collagen type IV that modulate the passage of TGF beta-1 and TGF beta-2 through the BM (from the tears, epithelium, endothelium, and/or aqueous humor) and into the stroma [5,7,25]. Perlecan produces a high negative charge due to its three heparan sulfate side chains [24,26] and, therefore, generates a non-specific barrier to TGF beta-1 and TGF beta-2 passage through the EBM (or DBM) into the stroma [5]. Collagen type IV directly binds TGF beta-1 or TGF beta-2 [2,27,28]; therefore, EBM (or DBM) collagen type IV also impedes the movement of the TGF betas through the BM into the corneal stroma [5,25]. Nidogen-1 and nidogen-2 in the EBM [5,25] also bind PDGF [29], and thereby modulate the transition of keratocytes to both corneal fibroblasts and myofibroblasts [30].

Once the EBM regenerates, TGF beta levels in the anterior stroma decline and precursor cells in transition to myofibroblasts either undergo apoptosis or, in the case of corneal fibroblasts, can revert to keratocytes [5,25]. Thus, the progression to myofibroblast-mediated fibrosis is halted. The relatively small amounts of disorganized ECM components, such as collagen type I, secreted by the corneal fibroblasts, are subsequently reorganized and/or phagocytized [31] by keratocytes, thereby returning the cornea to transparency [5,25].

### 2.4. Major Injuries and Fibrotic Healing in the Anterior Cornea

More severe injuries to the anterior cornea, such as chemical burns, lacerations, severe trauma, microbial infections, or laser vision correction PRK for high myopia (without intraoperative topical mitomycin C), commonly heal with significant stromal opacity (Figure 1C,E,F) and the generation of myofibroblasts and stromal fibrosis (Figure 6C–F) [5,25,32,33]. In these more severe injuries to the anterior cornea, even if the epithelium closes, the EBM is not fully regenerated in a timely manner (Figure 3C), with defective incorporation of perlecan being the best-characterized abnormality (Figure 7). Therefore, TGF beta-1 and TGF beta-2 penetrate the stroma to persistent levels (Figure 5D,E) adequate to drive the development of myofibroblasts (Figure 6C–E) from precursor corneal fibroblasts and fibrocytes [12,21,33].

There is a breakdown in the repair of the EBM in these corneas likely because the initial wave of keratocyte apoptosis and/or necrosis elicited by the injury is sufficiently large (Figure 6B) [15,18] that there are insufficient numbers of proximate keratocytes and corneal fibroblasts to coordinate the repair with the epithelium through the contribution of perlecan, nidogens, and collagen type IV [5,25,33]. Many of these injuries also produce severe anterior stromal surface irregularity that mechanically impedes EBM regeneration [34]. There can be other yet unrecognized factors in the cornea that inhibit the full regeneration of EBM, with lamina lucida and lamina densa, signaling maturity of the EBM [35].

Fibrosis must not be thought of only in terms of pathology. Clearly, the process serves an important protective function to maintain morphology in the organs where it develops, at least until excessive fibrosis leads to an advanced compromise of organ function. A good example of this principle is the corneal response to *Pseudomonas aeruginosa* keratitis (Figure 6E) [32]. If fibrosis did not rapidly develop in this quickly progressing and severe infection of the cornea, then perforation of the cornea and loss of the eye would occur much more frequently than is observed.

Even with severe full-thickness fibrosis of the corneal stroma caused by trauma or infection, there can be a surprising return of transparency and function [32,36]. Typically, this resolution of fibrosis occurs over a period of many months to years as the EBM and DBM are regenerated and myofibroblasts that are deprived of their ongoing, requisite supply of TGF beta undergo apoptosis [7,32]. Thus, the EBM can eventually be fully regenerated as keratocytes and/or corneal fibroblasts penetrate the fibrosis and cooperate with the corneal epithelium to restore a mature EBM [32,33]. Along with restoration of the normal epithelial barrier function (Figure 5C) [5], the mature EBM markedly diminishes the passage of TGF beta-1 and TGF beta-2 into the corneal stroma from the tears and corneal epithelium, and triggers myofibroblast apoptosis [37]. At this point, corneal fibroblasts and keratocytes repopulate that fibrotic stroma and re-establish the normal ultrastructure of the stroma associated with transparency by phagocytosis and reorganization [31], and in some cases eventually return the stroma to full transparency [5,25]. Essentially, the corneal fibroblasts and keratocytes clean up the disorganized ECM mess produced by the myofibroblasts.

Two or more potential myofibroblast precursor cells have been reported in well-studied organs, including skin, lung, and cornea [21,38,39,40]. The best characterized corneal myofibroblast precursors are fibroblasts derived from keratocytes and bone marrow-derived fibrocytes [21,22,30]. Epithelial to mesenchymal transition (EMT) and endothelial to mesenchymal transition (EndoMT) leading to myofibroblast development have not been well-characterized in the cornea. An in vitro study with corneal stromal and bone marrow (BM)-derived cells found that the numbers of SMA+ myofibroblasts generated from either keratocyte-derived precursor cells or BM-derived precursors were highest when both precursors were co-cultured in the same culture flask (juxtacrine), as when the two precursor cells were co-culture in different compartments of a Transwell System (paracrine) [41]. This suggests that the two different myofibroblasts cells potentiate the overall fibrosis response when they are present together in the corneal stroma. A proteomic study of corneal fibroblast-derived myofibroblasts compared with bone marrow-derived myofibroblasts found that 29% of proteins were differentially expressed between these two myofibroblast types [42], including proteins that contribute to the structure of fibrotic tissue, such as collagen types III, VII, and XI. Clues to progenitor-dependent differences in myofibroblasts were suggested by bioinformatic analysis of the differentially expressed proteins in that study [42]. Thus, canonical pathways involving oxidative phosphorylation, mitochondrial dysfunction, and sirtuin signaling were predominant in cornea-derived myofibroblasts, whereas pathways involving integrin signaling, glycolysis I, and remodeling of epithelial adherens junctions were predominant in BM-derived myofibroblasts. The Ingenuity Pathway Analysis of the differentially expressed proteins in these two myofibroblasts were also different, suggesting molecular and cellular functional differences [42]. Since BM-derived myofibroblasts produced much more collagen type XI and collagen type III, they likely contributed greatly to structure and strength of the fibrotic tissue in the cornea. Alternatively, since corneal keratocyte-derived myofibroblasts produced more collagen type VII, they more likely modulated cytokine production by adjacent fibroblasts in the healing stroma [42]. Thus, myofibroblasts derived from different precursors in a fibrotic tissue should not be thought of as equivalent, but rather as cells with similar phenotypes that contribute differentially to enhance the overall fibrosis response.

### 2.5. Injuries and Fibrotic Healing in the Posterior Cornea

The processes involved in the development of posterior corneal fibrosis involving DBM and the corneal endothelium (Figure 1G,H) [6,7] parallel those involving the EBM and the corneal epithelium in the anterior cornea [5,11]. Thus, injury and delayed regeneration of DBM leads to the penetration of high levels of TGF beta-1 and TGF beta-2 into the posterior stroma, although the primary sources of the fibrotic growth factors after posterior injury are from the aqueous humor and residual corneal endothelial cells [7]. Similarly, the precursor cells to myofibroblasts in the posterior cornea are fibroblasts derived from keratocytes and bone marrow-derived fibrocytes (Figure 6F–H) [6,7]. However, regeneration of DBM, if it occurs at all, tends to occur over a much longer period, measured in many months or years after injury, than regeneration of the EBM after its injury [7]. Thus, posterior corneal fibrosis tends to persist without corneal transplantation, especially in adult humans where there is limited endothelial proliferation in the absence of pharmacological manipulation [10].

## 3. Other Candidate Organs Where BM Injury Can Be Associated with Fibrosis

### 3.1. Skin

The skin BM (Figure 8A) [43] that separates the keratinized squamous epithelium from the underlying dermis has obvious parallels to the cornea. However, skin as an organ is exceedingly more complex than the cornea because of the accessory organs, such as hair follicles and sebaceous glands, as well as the vascularization of the dermis. This complexity is likely the explanation for why studies that parallel those for corneal fibrosis, for example after mechanical scrape injury, have not been reported for skin where unambiguous identification of cell phenotypes can be problematic. Nonetheless, there are numerous parallels to skin fibrosis caused by traumatic and thermal injuries [44] and in the skin manifestations of scleroderma [45].

Although there is some disagreement between different studies, that are likely related to antibody differences, keratinocytes produce TGF beta-1 and TGF beta-2 [46,47,48]. Other sources of skin TGF beta likely include bone marrow-derived cells in the dermal blood vessels and dermis, including monocytes and macrophages [49], and dermal fibroblasts themselves [50].

Skin has many potential precursors to myofibroblasts in fibrosis due to trauma and burns, as well as diseases such as scleroderma. These include dermal fibroblasts [51], keratinocytes via EMT [52,53], adipocytes [54], as well as pericytes [55] and fibrocytes [56] that migrate from the dermal blood vessels.

These parallels with the cornea, and similarities in anatomy and injuries, suggest that defective BM regeneration after skin injuries can have a role in dermal myofibroblast development and skin fibrosis. It would be of interest to determine if traumatic and thermal skin fibrosis is associated with defective perlecan incorporation into the keratinocyte BM similar to the cornea [5,25].

### 3.2. Lung

In some ways, the monolayer of alveolar epithelial type I cells overlying the alveolar BM (Figure 8B) and underlying interstitial space in lung alveoli is similar in organization to that of the corneal endothelial cells, Descemet’s membrane, and corneal stroma. Many toxic agents associated with idiopathic pulmonary fibrosis (IPF) (Figure 8C,D) and other fibrotic lung pathologies, such as tobacco smoke, bleomycin, paraquat, and butylated hydroxytoluene, produce chronic injury to the alveolar epithelial type I and II cells, and likely injury to the underlying BM [57,58]. Although there has been limited direct study of the ultrastructure and composition of the alveolar BM in these conditions, ultrastructural abnormalities, breaks, and convolution of the alveolar BM were clearly noted in transmission electron microscopic studies of IPF and other fibrotic lung diseases [57,58]. Fibrosis in interstitial lung diseases has been classically identified as fibrous tissue accumulation in the pulmonary interstitium within the alveolar walls bounded by the alveolar epithelial and capillary endothelial BMs [57]. Bowden [59] pointed out that insults that delayed the regeneration or interfere with the continuity between alveolar epithelial cells predispose to the development of pulmonary fibrosis. He also noted that delayed regeneration of the endothelial cells within the alveolus after some injuries, such as irradiation or butylated hydroxytoluene, also led to the accumulation of fibrotic myofibroblast cells [59]. Alveolar epithelial cell or alveolar endothelial cell injury is likely associated with injury and/or abnormal maintenance of the associated BMs which would likely alter BM regulation of TGF beta localization in these disorders.

The alveolar BM and endothelial BM in lung, similar to the corneal BMs and BMs in other organs, are composed of laminins, nidogens, perlecan, collagen type IV, and other components, some of which are tissue-specific [60,61]. These studies have shown that alveolar epithelial type I cells and endothelial cells produce these lung BM components, but other lung cells may also make contributions.

Several potential progenitors to myofibroblasts in lung fibrotic diseases have been supported by studies. A huge body of data was generated related to the alveolar epithelial type I or II cells themselves being myofibroblast progenitors via EMT in which TGF beta and/or other growth factors or cytokines trigger a major change in phenotype of the epithelial cells to mesenchymal myofibroblasts [62,63,64,65]. Kim and coworkers [66] engineered mice expressing the marker beta-galactosidase (beta-gal) exclusively in lung epithelial cells and then transiently overexpressed active TGF beta-1 in the lungs in vivo using an adenoviral vector (adTGF-β1) administered intranasally versus vehicle. After 21 days, lung sections revealed moderate fibrosis in the adTGF-β1-treated, but not vehicle-treated, mice. Clusters of x-gal-positive cells were noted within areas of lung with collagen deposition and some of the x-gal-positive cells were also alpha-smooth muscle actin-positive, supporting the EMT process occurring in vivo. Cell lineage tracing studies, however, raised questions about the importance of EMT in pulmonary fibrosis [67,68,69]. Therefore, EMT as a major source of myofibroblasts in pulmonary fibrosis remains controversial. Bone marrow-derived, blood-borne fibrocytes have been shown in several studies to have an important role in the generation of myofibroblasts in pulmonary fibrosis [70,71,72,73]. Alveolar septal fibroblasts have long been seen as likely progenitor cells to myofibroblasts [67,74]. One study found that pericytes were an important progenitor to myofibroblasts in fibrotic lungs [69], but that remains controversial [67]. Finally, there is a possibility that endothelial to mesenchymal transition (EndoMT) can have a role in some fibrotic lung diseases and; therefore, vascular endothelial cells can serve as the progenitor cells in these lung diseases [75]. Likely, as in the cornea, there are several progenitors to myofibroblasts in fibrotic lungs and these myofibroblasts may have differing functions in the fibrosis response [42].

There is an old maxim in criminal and civil litigation that can be briefly summarized as “follow the money.” In fibrosis research, the analogous maxim is “follow the TGF beta” because without excessive production or activation, or anomalous localization, of TGF beta, it is unlikely fibrosis will develop in a tissue. In the lung, there are several potential sources of TGF beta that have been associated with fibrotic lung diseases, many of these are associated with chronic injury to the alveolar epithelium [76]. These sources include alveolar macrophages, neutrophils, eosinophils, endothelial cells, fibroblasts, “activated alveolar epithelial cells,” and even the myofibroblasts themselves once they develop in lung tissues [76,77,78,79,80,81,82]. Type II alveolar epithelial cells and interstitial fibroblasts were also shown to express connective tissue growth factor (CTGF), a growth factor associated with fibrosis in which the transcription is activated by TGF beta, in IPF [83]. Many of the myriad activators of latent TGF beta are present in healthy and fibrotic lung tissues and the expression and localization of these TGF beta activators is likely important in the pathophysiology of many fibrotic lung diseases [2,84].

It seems likely that chronic injury to the alveolar epithelium [76] would also lead to damage and possibly altered composition of the associated alveolar BM, although this has been little studied. One study [85] found in a bleomycin model of fibrosis in hamsters that there was focal injury to the alveolar epithelial cells and the alveolar epithelial BM associated with acute inflammation by 6 days after bleomycin exposure. The BM damage included denudation and thickening of the alveolar epithelial BM. By 60 days after exposure, although the alveolar epithelium had regenerated, there remained BM abnormalities of thickening and duplication of the alveolar epithelial BM that was most prominent in the fibrotic areas of the lung. No alterations in the capillary endothelial BM were noted in this model. Studies such as these can be especially revealing if they included time course experiments after injury with multiplex immunohistochemistry for BM components such as perlecan, collagen type IV, nidogens, and laminins, similar to studies performed in the cornea [5,25].

In vitro studies of alveolar BMs have found using rat alveolar type II cells transfected with the SV40-large T antigen gene, to induce extended life of the cells, that were then propagated on type I collagen matrix gels [86]. Only when pulmonary fibroblasts were present in the gel did the alveolar cells produce a thin continuous BM. These alveolar BMs contained the typical BM components, including the perlecan and collagen type IV modulators of TGF beta localization by BMs [26,27,28,29], as well as laminins and nidogens. In that study, pulmonary fibroblasts supplied soluble components to the generating BM [86], similar to what was found for keratocyte/corneal fibroblast contributions to regeneration of the epithelial BM in corneas [5,25]. Another group confirmed the importance of alveolar epithelial-pulmonary fibroblast interactions in the generation of the alveolar epithelial BM in a similar in vitro mouse model [87].

These similarities to BM changes in corneas suggest that the alveolar BM has a role in modulating alveolar macrophage or other cellular TGF beta localization into the acinar interstitial spaces to modulate myofibroblast development from septal fibroblasts and fibrocytes in conditions where there are chronic insults to acinar epithelial cells, for example, caused by smoke, bleomycin, paraquat, butylated hydroxytoluene, and other agents.

## 4. Liver Fibrosis: Capillarization of Hepatic Sinusoids Associated with the Generation of Endothelial BMs

The liver is a structurally unique organ where BMs do not have a role in normal physiology, but the appearance of BMs likely contributes to the pathophysiology of fibrosis. This is because the distinctive organization of hepatic tissue necessary for its functions requires direct cellular contact to perform detoxification, modification, and excretion of endogenous and exogenous substances, including toxins. Thus, there are no BMs associated with hepatocytes, endothelial cells, vascular channel sinusoids or the spaces of Dissé in normal liver (Figure 9). Hepatic stellate cells (HSCs) exist in a quiescent state within this complex network of extracellular matrix in the space of Dissé. HSCs, previously called vitamin A-storing cells, lipocytes, interstitial cells, fat-storing cells, or Ito cells, however, secrete laminins, proteoglycans (including perlecan) and collagens (including collagen type IV) to contribute to the local extracellular matrix. HSCs are normally important storers of vitamin A [88,89,90]. Nidogen-1 and nidogen-2 were also detected in precursors to HSCs during liver development in mice using in situ hybridization [91]. It is interesting that four major components of BMs are produced but no traditional BMs are detected using TEM in proximity to hepatocytes, endothelial cells, the vascular channels of the sinusoids, or the spaces of Dissé in normal liver. Presumably, this is because the makeup of this ECM in the space of Dissé does not include properly localized, self-polymerizing laminins required to initiate BM formation [24], does not have the appropriate stoichiometry of BM components for BM assembly, or contains other components that actively inhibit BM formation.

A recent review by Karsdal et al. [92] emphasized ECM changes that occur in liver fibrosis related to BM components and the interstitial matrix (IM) and how they are different depending on the etiology of the injury. For example, the fibrosis pattern of early-stage chronic viral hepatitis is characterized as a periportal fibrosis with increased interstitial collagens and dense peribiliary BMs [92]. Conversely, fibrosis due to alcoholic or nonalcoholic steatohepatitis (NASH) is characterized by pericellular accumulation of BM proteins and production of small amounts of collagen type III and other fibrillar collagens by HSCs. Increased collagen type IV is the first sign of early fibrosis in NASH [92].

In many chronic liver diseases, a pathological finding often noted is what is referred to as “capillarization of hepatic sinusoids” [93]. This includes the formation of extraneous BM beneath the endothelial cells of the sinusoids, the loss of the normal endothelial fenestrations (defenestration) (Figure 9) and the transformation of sinusoidal endothelium to a more vascular type of endothelium. These pathophysiological changes are thought to interfere with the exchange of materials between the sinusoidal blood and the hepatocytes [94]. Thus, in liver the generation of sinusoidal BMs in fibrosis is detrimental to liver function. Capillarization of hepatic sinusoids is non-specific and can occur with alcoholic liver fibrosis, autoimmune hepatitis, and primary biliary cirrhosis in humans and in animal models it can be induced in hepatic fibrosis models triggered by dimethylnitrosamine, carbon tetrachloride, and selenium [93]. Capillarization is accompanied by an increase in collagen type IV and collagen type XVIII content within the space of Dissé [95,96]. A similar abnormality, that is termed “pseudocapillarization of sinusoids,” has been noted in livers of the elderly without fibrosis [97,98].

What is the ongoing source of TGF beta that drives liver fibrosis? Most etiologies for fibrosis, such as viral infections, autoimmune disorders, allergic diseases, and toxic exposures, are associated with chronic inflammation and many investigations have supported macrophages as the primary source of TGF beta-1 that drives fibrosis [79,99,100]. However, this remains controversial since other cells, including Kupffer cells, liver sinusoidal endothelial cells, resident dendritic cells, and even hepatic stellate cells, have been shown to produce TGF beta-1 [101]. In one study, platelet TGF beta-1 was found to have an important role in liver fibrosis induced by carbon tetrachloride in mice [102]. It is possible that which cells produce the pro-fibrotic TGF beta-1 (and TGF beta-2) depends on the specific liver disorder and the stage of development of fibrosis.

Similarly, it remains controversial which cells give rise to myofibroblasts in liver fibrosis. Many studies in liver fibrosis of varying etiologies have focused on HSCs being the main precursor cells of myofibroblasts that populate the organ during fibrosis [90,99,103,104]. However, in a study that used lineage fate tracing methods, Mederacke and coworkers [104] concluded that while HSCs were the dominant progenitor to myofibroblasts regardless of the etiology of liver fibrosis, there were other precursors as well. Thus, there is also evidence that bone marrow-derived fibrocytes generate liver myofibroblasts [105,106]. There is also evidence, albeit controversial, that hepatocytes or cholangiocytes via EMT [107] and sinusoidal endothelial cells via EndoMT [108] can serve as progenitors to myofibroblasts. What other precursors besides HSCs contribute to fibrosis may depend on the specific etiology of the liver injury. Regardless of the precursor, however, it does not appear that traditional BMs have a role in modulating TGF beta or other pro-fibrotic regulators in liver fibrosis. It is possible that collagen type IV, because it directly binds TGF beta-1 [2,27,28], as well as perlecan [26] or other molecules, can modulate liver TGF beta within the microenvironment of the space of Dissé, but not via traditional BMs. This can be similar to corneal fibroblasts producing collagen type IV in the corneal stroma far from the BMs to modulate TGF beta [7]. However, in the cornea, it is not known whether the collagen type IV detected by immunohistochemistry in the stroma far away from BMs after fibrotic injuries is full-length collagen type IV or rather degradation fragments of collagen type IV. Collagen type IV degradation fragments, such as arresten (alpha-1 chain) and canstatin (alpha-2 chain), can have important functions, such as inhibition of neovascularization.

## 5. Conclusions

The fibrosis response, to a wide range of injuries, is of obvious importance in virtually every organ where it was investigated. It seems unlikely that this overall process would be unique to each individual organ. Rather, it seems probable that the systems in place to promote fibrosis, and its resolution, would be generally utilized throughout the organism, except possibly in organs with specialized functions, such as the liver, that require a structure free of traditional BMs. With that in mind, this author is of the opinion that the importance of the BMs in corneal fibrosis from traumatic, infectious, chemical, and surgical injuries, where it is most easily studied without numerous potentially confounding cells, is likely to be also relevant in the many other organs where fibrosis is important in the pathophysiological response to injuries and diseases.

## Figures and Tables

**Figure 1 cells-11-00309-f001:**
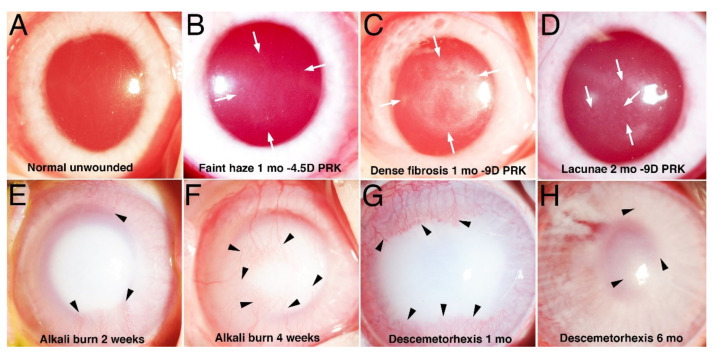
Slit lamp photographs of haze and scarring fibrosis in rabbit corneas. (**A**) Normal unwounded transparent cornea. (**B**) One month after −4.5D PRK a cornea has faint opacity (haze) within arrows [5]. (**C**) One month after −9D PRK a cornea has dense scarring fibrosis within arrows [5]. (**D**) At 2 mo. after −9D PRK areas of clearing (lacunae, arrows) are developing within scarring fibrosis [5]. (**E**) Dense scarring fibrosis 2 weeks after 5 mm surface alkali burn with 1 N NaOH. Stromal neovascularization (arrowheads) begins to develop. (**F**) Scarring fibrosis has progressed at 4 weeks after alkali burn. Stromal neovascularization (arrowheads) has progressed. (**G**) Dense scarring fibrosis 1 mo. after 8mm Descemetorhexis. Stromal neovascularization (arrowheads) has developed [7]. (**H**) Scarring fibrosis has diminished by 6 mo. after Descemetorhexis with iris details now visible. Most of the opacity that remains is associated with the corneal neovascularization (arrowheads) [7]. Mag. 20×.

**Figure 2 cells-11-00309-f002:**
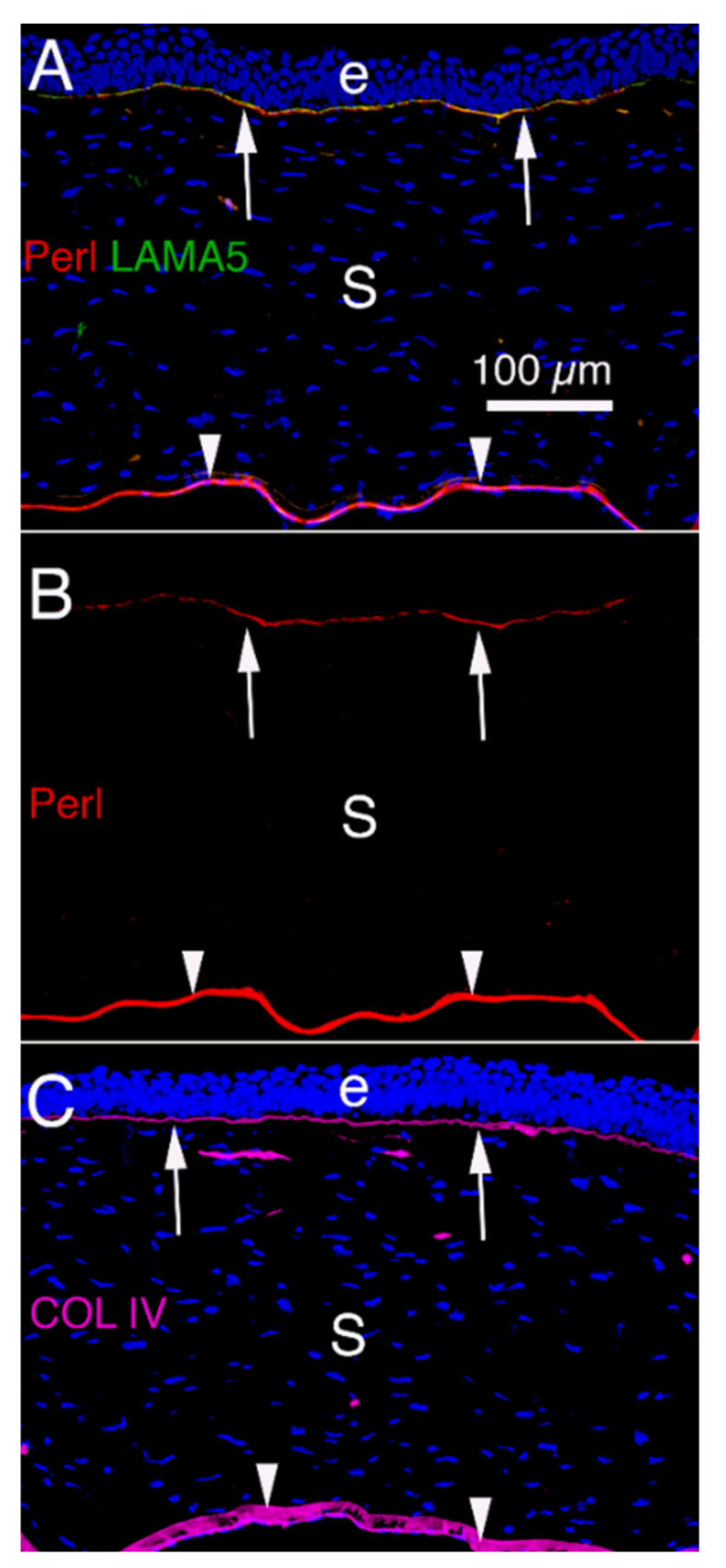
Corneal BM components that modulate TGF beta-driven myofibroblast development and fibrosis in unwounded rabbit corneas [5]. (**A**) Immunohistochemistry (IHC) for perlecan (Perl), as well as laminin alpha-5 (LAMA5) [5]. (**B**) IHC for perlecan alone. (**C**) IHC for collagen type IV. Arrows indicate the EBM with overlying epithelium (e) and arrowheads indicate Descemet’s membrane that overlies the corneal endothelium, respectively, in all panels. S is stroma populated primarily with keratocytes. Blue is DAPI stained nuclei. Mag. 200×.

**Figure 3 cells-11-00309-f003:**
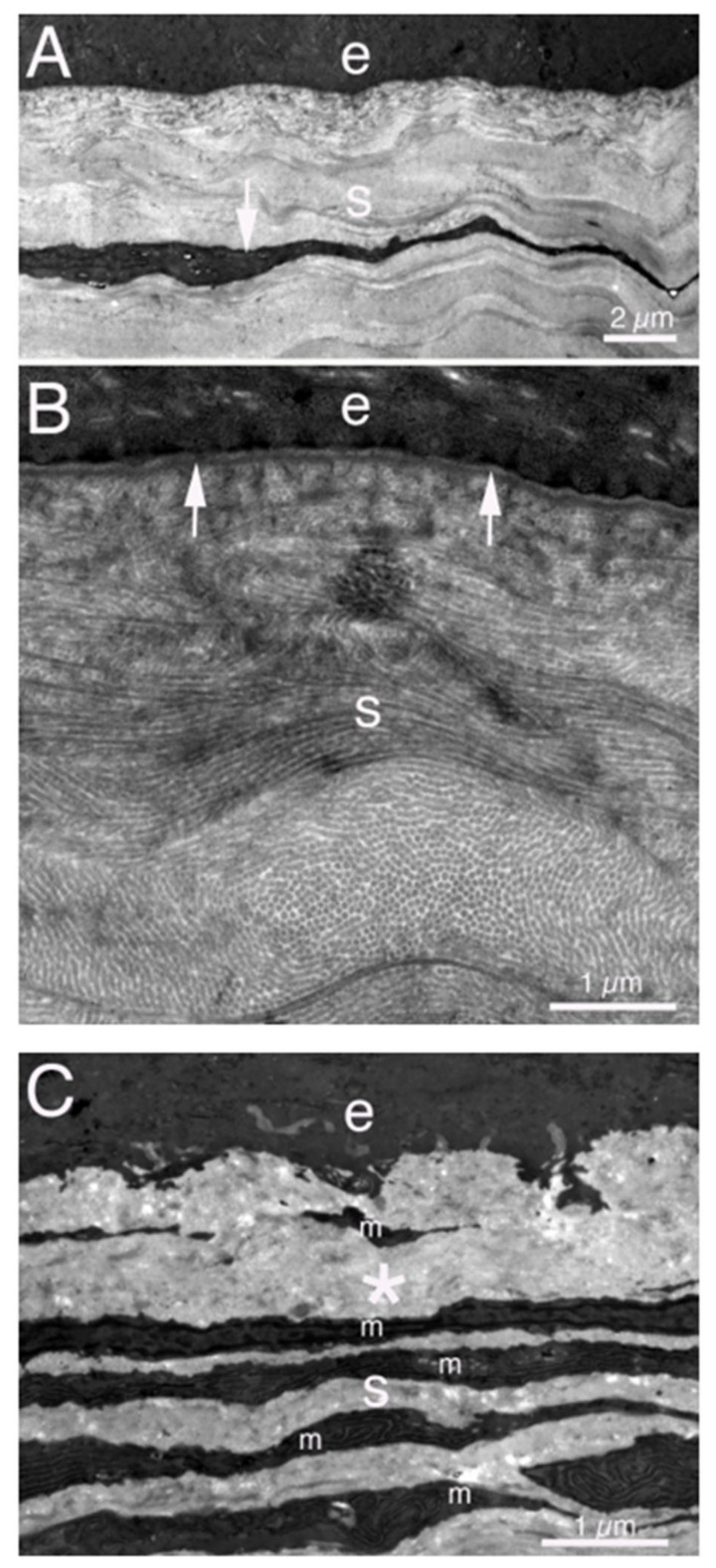
TEM of normal and fibrotic rabbit corneas. (**A**) Lower magnification image of an unwounded cornea showing the epithelium (e) and stroma (s) with a keratocyte (arrow). (**B**) Higher magnification image showing the epithelium (e) with the underlying EBM. The arrows indicate the lamina lucida anterior to the lamina densa of the EBM. In the stroma (s) note the uniform diameter of the collagen fibrils, with some seen in cross-section and others longitudinally, and the highly ordered packing of the fibrils. (**C**) In a cornea with severe fibrosis at 1 month after PRK, the stromal ECM is highly disorganized (*), without evidence of regular fibrils or packing. The anterior stroma (S) is also populated with many layered myofibroblasts (m). These images were previously unpublished but from the study of Torricelli et al., Investig. Ophthalmol. Vis. Sci. 2013, *54*, 4026–4033.

**Figure 4 cells-11-00309-f004:**
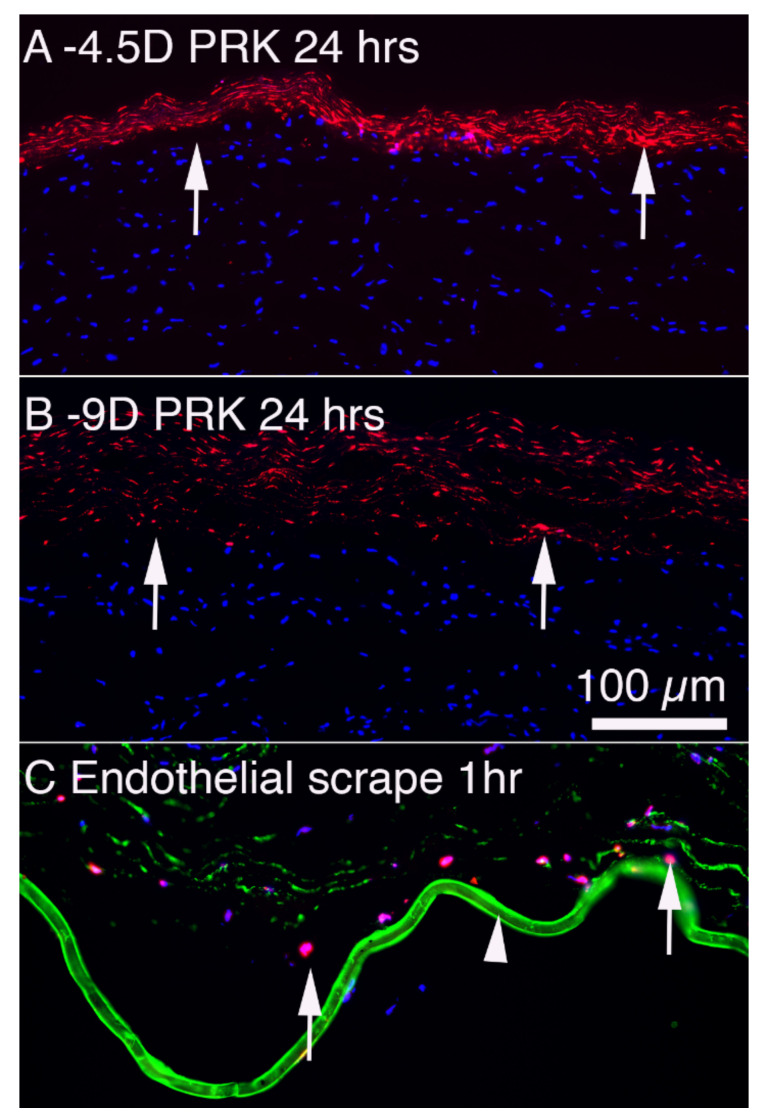
Keratocyte apoptosis in response to injury in rabbit corneas. (**A**) TUNEL assay at 24 h after −4.5D PRK that entails epithelial debridement and then anterior stromal ablation with the excimer laser. Arrows indicate anterior stromal keratocytes undergoing apoptosis. The apoptotic cells can be detected with TEM within moments of epithelial scrape but become strongly TUNEL+ from 4 to more than 24 h. Many bone marrow-derived cells such as monocytes and fibrocytes detected with markers such as CD34, CD45, and CD11b infiltrate the stroma from the limbus and many also undergo apoptosis in the first 24 to 72 h. (**B**) At 24 h after −9D PRK, with twice the number of excimer laser pulses, many more anterior stromal keratocytes (arrows) undergo apoptosis. Thus, there is a correlation between the magnitude of the anterior corneal injury and the number of keratocytes that undergo early apoptosis [19]. (**C**) At 1 h following an 8 mm corneal endothelial scrape injury, many posterior stromal keratocytes (arrows) undergo apoptosis detected with the TUNEL assay. Note the edema of the stroma that also occurs immediately after endothelial injury. The arrowhead indicates DBM stained (green) for BM component nidogen-1. Figures (**A**,**B**) were previously unpublished but from the study of Mohan et al., 2003 [19]. Figure (**C**) reprinted with permission from Medeiros et al. Exp. Eye Res. 2018; 172:30-35.

**Figure 5 cells-11-00309-f005:**
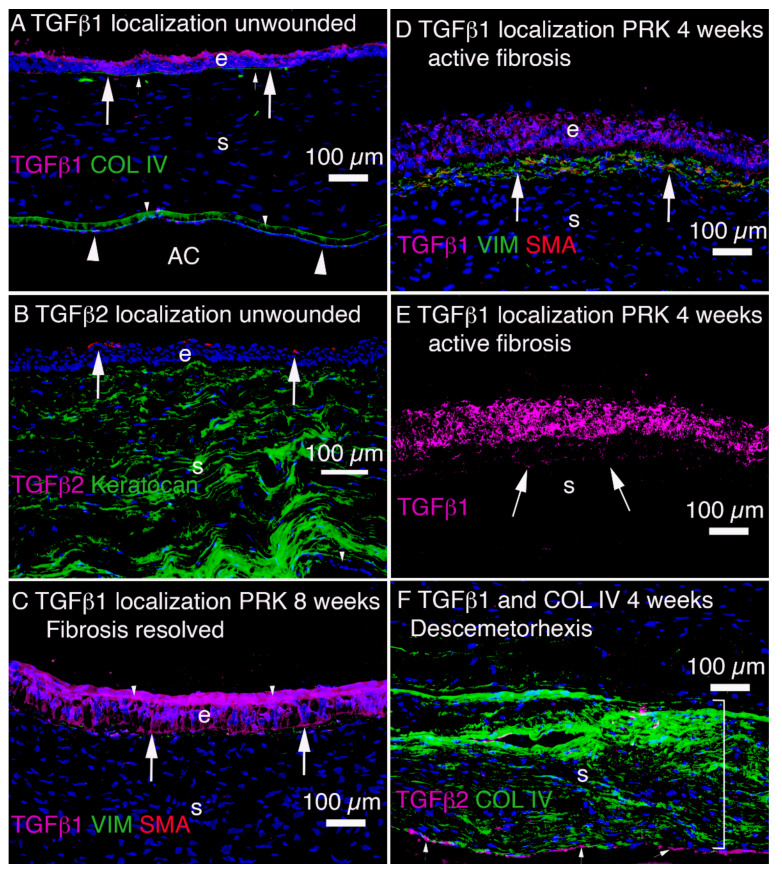
Localization of TGF beta-1 and TGF beta-2 in unwounded and wounded rabbit corneas. (**A**) TGF beta-1 (TGFb1) is produced (large arrows) in corneal epithelium (e) and endothelium (large arrowheads) and is also present in tears and aqueous humor in the anterior chamber (AC) [5]. In unwounded cornea, collagen type IV (COL IV) is detected in the EBM (small arrows) and in DBM (small arrowheads). (**B**) TGFb2 is not expressed in the corneal epithelium or corneal endothelium (arrowhead indicates a small area of visible corneal endothelium) but is present in tears (produced by accessory and main lacrimal glands) and in the aqueous humor. (**C**) In corneas that do not develop fibrosis or in corneas that develop fibrosis that subsequently resolves, as in this cornea at 8 weeks after PRK, TGFb1 is retained from entering the stroma by the fully regenerated EBM (arrows) and regeneration of the superficial epithelial barrier function (EBF, small arrowheads). Note no SMA-positive myofibroblasts remain, but a few vimentin-positive, SMA-negative corneal fibroblasts persist just posterior to the EBM. (**D**) In a cornea that develops fibrosis 4 weeks after PRK, high levels of TGFb1 (and TGFb2 not shown) accumulate throughout the epithelium (e) and into the anterior stroma (s) without evidence of EBM regeneration or EBF. Numerous SMA-positive myofibroblasts (arrows) and SMA-negative, vimentin-positive corneal fibroblasts are present in the sub-epithelial stroma. (**E**) The same section as in D, but showing only TGFb1, highlights the penetration of the TGFb1 into the anterior stroma (arrows), although some stromal cells also produce limited amounts of TGFb1 [5]. (**F**) In a rabbit cornea at 4 weeks after Descemetorhexis removal of an 8 mm disc of DBM and corneal endothelium, TGFb1 (arrows) is localized at the posterior corneal surface still devoid of DBM or endothelium. Much of the posterior stroma (bracket) contains collagen type IV (COL IV) not associated with DBM that is upregulated in corneal fibroblasts by TGFb1. Since COL IV directly binds TGFb1 in competition with cognate TGF beta receptors, it is hypothesized this COL IV is produced to downregulate TGFb1 effects on cells in the posterior stroma, including myofibroblast precursors [7]. A similar upregulation of non-EBM COL IV occurs in the anterior stroma after injuries such as PRK. Panels A and B are previously unpublished images from study de Oliveira et al. Exp Eye Res, 2021;202:108325. Panels C, D and E reprinted with permission from de Oliveira et al. Exp Eye Res, 2021;202:108325. Panel F reprinted with permission from Sampaio LP et al. Exp Eye Res. 2021;213:108803.

**Figure 6 cells-11-00309-f006:**
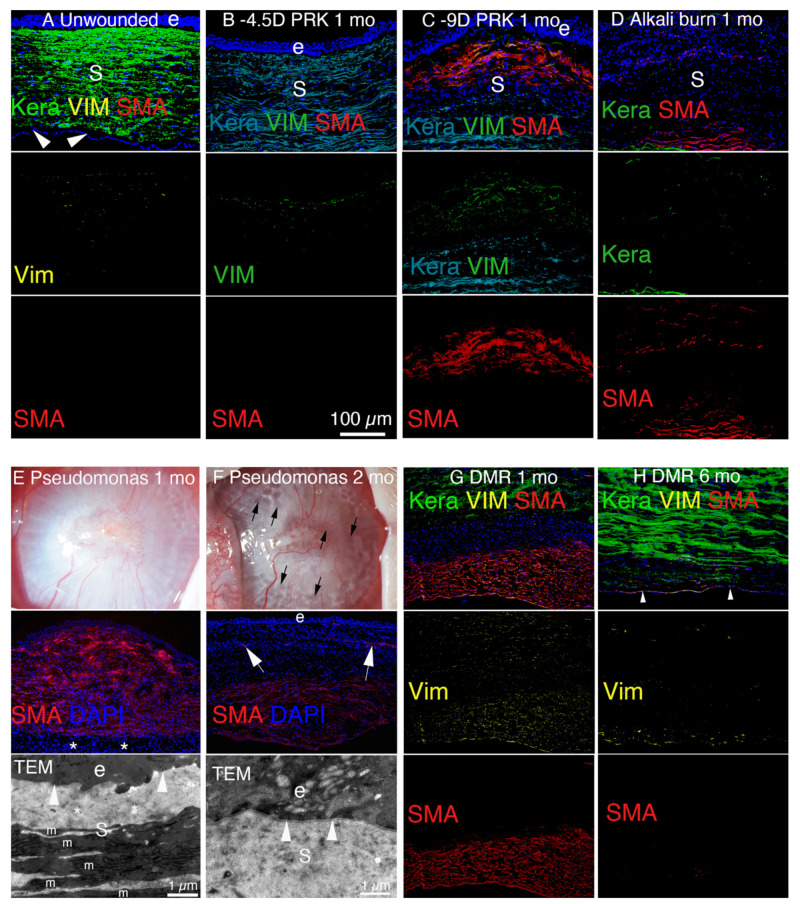
Stromal cellularity of a normal cornea and corneas after injuries that heal without fibrosis and with fibrosis in rabbits. (**A**) The unwounded corneal stroma (s) is populated with keratocan-positive keratocytes between the epithelium (e) and corneal endothelium (arrowheads). At the vimentin (vim) antibody concentration used [5,25], only a few anterior stromal keratocytes were vimentin positive. No SMA-positive cells were detected. (**B**) One month after −4.5D PRK, there were numerous vimentin-positive corneal fibroblasts in the anterior stroma but the stroma was mostly populated with keratocan-positive keratocytes. No SMA-positive myofibroblasts were detected. (**C**) One month after −9D PRK, the anterior stroma is populated with SMA-positive, vimentin-positive myofibroblasts and SMA-negative, vimentin-positive corneal fibroblasts (and possibly fibrocyte progeny). (**D**) At 1 month after a one-minute 1N NaOH alkali burn that also destroyed the endothelium and Descemet’s membrane, the full-thickness corneal stroma is filled with myofibroblasts and corneal fibroblasts. Few keratocytes are detected. (**E**) At 1 month after infection with Pseudomonas aeruginosa keratitis sterilized with topical tobramycin there is severe opacity of the cornea in a slit lamp photograph. In IHC, approximately 90% of the stroma is filled with SMA-positive myofibroblasts, and in this cornea sparred only the most posterior stroma adjacent to the corneal endothelium. In a TEM image of this cornea, no lamina lucida/lamina densa is detected. The stroma (s) has disorganized ECM (*) and numerous myofibroblasts (m). (**F**) At 1 month after infection with Pseudomonas aeruginosa keratitis sterilized with topical tobramycin, the opacity in the cornea decreases and numerous transparent areas called lacunae (black arrows) develop. On IHC in this cornea where the Pseudomonas aeruginosa extended through the entire cornea and destroyed the corneal endothelium, SMA-positive myofibroblasts populate the posterior stroma but myofibroblasts disappeared in the anterior stroma. Corneal neovascularization (arrows) with SMA-positive pericytes develops. In a TEM image, lamina lucida/lamina densa (arrowhead) was regenerated. The stroma (s) had organized collagen fibrils similar to normal unwounded stroma and no myofibroblasts were detected in the anterior stroma. (**G**) At 1 month after Descemetorhexis (DMR), the posterior stroma is filled with SMA-positive myofibroblasts. The more anterior stroma in this section had keratocan-positive keratocytes. An intermediate layer of SMA-negative, keratocan-negative, vimentin-positive corneal fibroblasts (and likely fibrocyte progeny) are present between the keratocyte and myofibroblast layers. (**H**) At 6 months after DMR, the corneal endothelium (arrowheads) regenerates. Most of the posterior stroma is repopulated with keratocan-positive keratocytes, except adjacent to the corneal endothelium and regenerated DBM there were numerous keratocan-negative, SMA-negative, vimentin-positive corneal fibroblasts and a few remaining SMA-positive myofibroblasts. e is epithelium and s is stroma in all panels. Blue is DAPI-stained nuclei in all panels. Panels B and C reprinted with permission from de Oliveira et al. Exp Eye Res 2021:202;108325. Panels E and F reprinted with permission from Marino et al. Exp Eye Res. 2017;161:101-105. Panels G and H reprinted with permission from Sampaio LP et al. Exp Eye Res. 2021;213:108803.

**Figure 7 cells-11-00309-f007:**
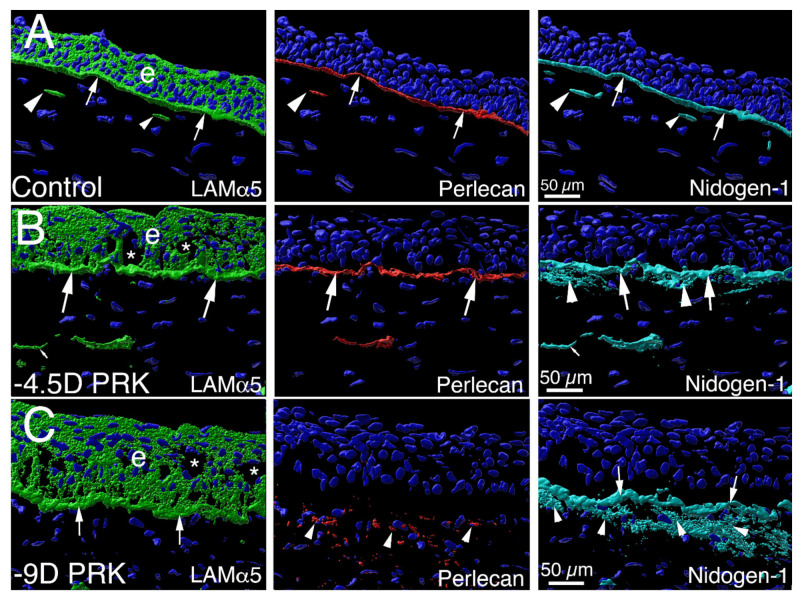
Defective perlecan EBM incorporation in a PRK injured rabbit cornea that developed scarring stromal fibrosis and myofibroblasts. Confocal microscopy Imaris 3D constructed images of triplex IHC for laminin alpha-5, perlecan and nidogen-1 in an unwounded control cornea and corneas with moderate −4.5D PRK and severe −9D PRK epithelial-stromal injury [5]. (**A**) Laminin alpha-5 (green) was detected in the epithelium (e) and in the EBM (arrows) in an unwounded cornea. Two DAPI-negative vesicles with laminin alpha-5 (arrowheads) are present in the anterior stroma adjacent to the EBM. These were likely produced by keratocytes to contribute to maintenance of the EBM. Perlecan (red) was detected in the EBM (arrows), and in vesicles in the anterior stroma (arrowhead). Nidogen-1 (blue gray) is a major component in the EBM (arrows) and is present in secretory vesicles in the anterior stroma (arrowheads). (**B**) A cornea at 1 month after surgery that had moderate epithelial-stromal injury (−4.5D PRK) and did not develop myofibroblasts or scarring stromal fibrosis (see Figure 1B). The laminin alpha-5, perlecan and nidogen-1 localization in the EBM are similar to that noted in the unwounded cornea (large arrows), except there are increased nidogen-1 (arrowheads) in the subepithelial stroma surrounding stromal keratocyte/corneal fibroblast cells. Vesicles (small arrows) that are DAPI-negative are present in the anterior stroma and contain one or more of the EBM components. (**C**) In a cornea 1 month after more severe epithelial-stromal injury (−9D PRK) there is greater stromal opacity and myofibroblasts (see Figure 1C). Laminin alpha-5 and nidogen-1 (arrows) EBM localization is similar to that noted in the unwounded control cornea. Perlecan, however, was not detected at significant levels in the EBM, even though it is present within and surrounding myofibroblasts (arrowheads) in the anterior stroma. Stromal nidogen-1 (arrowheads) surrounding myofibroblasts is also present at high levels in the anterior stroma. Blue in all panels is DAPI-stained nuclei. e is epithelium. * indicates artifactual defects in the epithelium which are often noted in PRK corneas that are cryo-sectioned in the first 1 to 2 months after surgery while the EBM has not fully regenerated. Reprinted with permission from de Oliveira et al. Exp Eye Res 2021:202;108325.

**Figure 8 cells-11-00309-f008:**
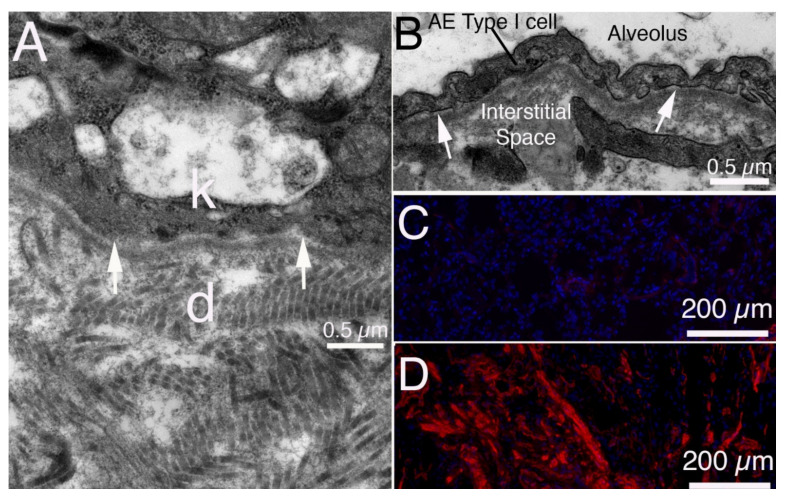
Organs where BMs can have a role in fibrosis [43]. (**A**) TEM in normal rabbit skin. The basal keratinocyte (k) and dermis (d) are separated by the BM with lamina lucida (arrows) and lamina densa. Note the larger and more disorganized fibrils in the dermis compared with corneal stroma in Figure 3b. (**B**) TEM in normal rabbit lung. The alveolar BM with lamina lucida (arrow) and underlying lamina densa separates the alveolar epithelial cell type I (AE cell type I) from the interstitial space. (**C**) IHC for SMA in normal human lung primarily stains pericytes associated with blood vessels. There is little staining for SMA in the normal lung parenchyma. Blue is DAPI stained nuclei. (**D**) In a human lung with advanced idiopathic pulmonary fibrosis (IPF) SMA-positive myofibroblasts are present throughout the parenchyma. Blue is DAPI stained nuclei.

**Figure 9 cells-11-00309-f009:**
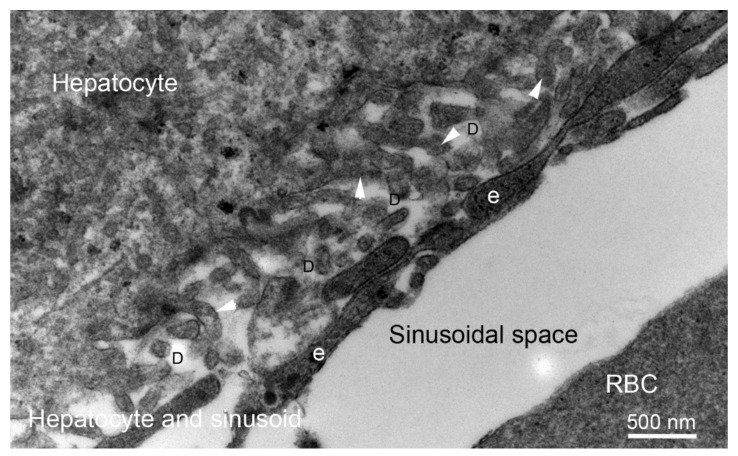
TEM of normal liver in the rabbit. No BM is associated with hepatocytes, endothelial cells (e), vascular channel sinusoids or spaces of Dissé (D). The hepatocyte has processes (arrowheads) that extend into the space of Dissé. The sinusoids have a discontinuous, highly fenestrated endothelial lining. Neither the hepatocytes nor endothelial cells have BMs that separate them from the space of Dissé. Mag. 30,000×. Reprinted with permission from Saikia et al. Cell and Tissue Res, 2018;374:439-453.

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
