# Peer review of "Fibrosis Is a Basement Membrane-Related Disease in the Cornea: Injury and Defective Regeneration of Basement Membranes May Underlie Fibrosis in Other Organs"

_cells, 2022, doi:10.3390/cells11020309_

Round 1
Reviewer 1 Report
I like the manuscript - and I could not wait to read it because of the title and my deep interests in fibrosis.
However, i feel a little mislead, because of the heavy focus on cornea (which suggests that cornea should be in the title). But I did like to read about the epithelial cell side of the story, that is translatable to skin, lung and liver - which to me is the central piece of the story. The common denominators.
I think the thoughts on collagen localization and ECM composition from this paper, "Hepatology, 2020 Jan;71(1):346-351, Is the Total Amount as Important as Localization and Type of Collagen in Liver Fibrosis Attributable to Steatohepatitis?" are central to the discussion, and possible a schematic figure on how structure, composition, and co-locations of the BM in relation to the interstitial (IM) ECM.
I would limit the cornea, include a schematic figure inspired from the above paper, discuss the main components of IM and BM in a small section, and emphasize the common denominators.
I truly do like the paper and would like to see a slightly revised version.
Lastly - it is a fine balance between a review and unpublished research. In figures, not all pictures have references to previous published papers, which suggest this is new or additional work from those papers. This needs to be carefully addressed.
If the sole authors wants, I will be happy to work with him to modify. This is a very fine paper, that deserves to be published.
Author Response
1. I feel a little mislead, because of the heavy focus on cornea (which suggests that cornea) should be in the title. But I did like to read about the epithelial cell side of the story, that is translatable to skin, lung and liver - which to me is the central piece of the story. The common denominators.
Corneal researchers interested in fibrosis are well-aware of the information provided in the cornea section of this paper. My goal here in this invited review for the Special Issue "Cellular and Molecular Mechanisms of Fibrosis" that Nina Zidar, MD, PhD asked me to prepare was to review the cornea work that conclusively demonstrates the critical role of the EBM and Descemet’s basement membrane in modulating fibrosis in the cornea for researchers in other fields and then highlight some examples where that model is likely to be relevant and then discuss the liver example where the absence of basement membranes associated with hepatocytes or the endothelial cells in the sinusoids might suggest this model is not relevant in liver, but really it is because of the collagen type IV and other components of BMs produced there. I changed the title of the article to “Fibrosis Is a Basement Membrane-Related Disease in the Cornea: Injury and Defective Regeneration of Basement Membranes May Underlie Fibrosis in Other Organs” with the second half a sub-title so it will be clear to readers what the article is about.
2. I think the thoughts on collagen localization and ECM composition from this paper, "Hepatology, 2020 Jan;71(1):346-351, Is the Total Amount as Important as Localization and Type of Collagen in Liver Fibrosis Attributable to Steatohepatitis?" are central to the discussion, and possible a schematic figure on how structure, composition, and co-locations of the BM in relation to the interstitial (IM) ECM.
Thank you for bringing this paper to my attention. It is highly relevant to the topic and I have now cited it and discussed it in the liver section of the paper lines 507-514. I’m confused about what a schematic figure should include. Are you meaning something like Fig. 2B and C in the Hepatology paper by Karsdal et al.? A little more description of the figure you envision would be very helpful.
3. I would limit the cornea, include a schematic figure inspired from the above paper, discuss the main components of IM and BM in a small section, and emphasize the common denominators.
As I mentioned above, a main objective of this invited paper was to highlight what has been discovered in the cornea for others interested in fibrosis that are unaware of the progress made in the cornea. I’m happy to include a schematic diagram from the paper but I am confused about specifically what is being asked. Again, do you mean something like Fig. 2B and C in the Hepatology paper by Karsdal et al.?
4. Lastly - it is a fine balance between a review and unpublished research. In figures, not all pictures have references to previous published papers, which suggest this is new or additional work from those papers. This needs to be carefully addressed.
All of the figures in this paper related to the cornea were from published studies. However, in each of these studies each group at each time point would have 4 to 6 corneas analyzed or had additional images from the same corneas but different from those in the previously published paper. Therefore, I used other images from these published studies, so I didn’t need to ask permission from the original publication for many of them. I did go back and make sure that the figure legends all include the reference. Some journals, for example Matrix Biology, do not want figures that have been previously published, and that is why I used alternative figures from these studies in this Cells submission in many of the cornea figures.
5. If the sole authors wants, I will be happy to work with him to modify. This is a very fine paper, that deserves to be published.
I’m happy to further edit the paper with specific suggestions. However, I want to make sure researchers in other organs can appreciate the cornea work that points to the importance of the BM in modulating fibrosis and injury to the BM with defective regeneration being a critical aspect of fibrosis in the cornea. If a schematic figure is needed please give me a little more guidance on that figure's design
Reviewer 2 Report
This manuscript describes that relationship between basement membrane injury and fibrotic disease. The author interestingly and appropriately explains fibrosis. However, there are several points that should be modified before publication.
- Title is “Fibrosis is often a basement membrane-related disease”. However, most of manuscript is about corneal fibrosis. It seems to need a title modification or a manuscript modification. Even the figures, 6 of 9 figures are about cornea.
- I felt this paper is not a review paper, because, in this manuscript, some figures (Figure 1-3, 8) were taken by the author, and some figures (Figure 4-7, 9) were already published elsewhere. I believe that a review paper should use summarized tables or figures which were made by author (by that I do not mean pathology or immunohistochemistry data. These are for original research paper). Therefore, I suggest to reorganize the paper.
- Author submitted the manuscript to “cellular molecular mechanisms fibrosis” section. This manuscript is very well-organized, however, the context of manuscript is too focused on a simple explanation. I believe that a review paper should draw new conclusions from the existing data.
Author Response
- Title is “Fibrosis is often a basement membrane-related disease”. However, most of manuscript is about corneal fibrosis. It seems to need a title modification or a manuscript modification. Even the figures, 6 of 9 figures are about cornea.
I changed the title of the article to “Fibrosis Is a Basement Membrane-Related Disease in the Cornea: Injury and Defective Regeneration of Basement Membranes May Underlie Fibrosis in Other Organs” with the second half a sub-title so it will be clear to readers what the article is about. My goal here in this invited review for the Special Issue "Cellular and Molecular Mechanisms of Fibrosis" that Nina Zidar, MD, PhD asked me to prepare was to review the cornea work that conclusively demonstrates the critical role of the EBM and Descemet’s basement membrane in modulating fibrosis in the cornea for researchers in other fields and then highlight some examples where that model is likely to be relevant and then discuss the liver example where the absence of basement membranes associated with hepatocytes or the endothelial cells in the sinusoids might suggest this model is not relevant in liver, but really it is because of the collagen type IV and other components of BMs produced there.
2. I felt this paper is not a review paper, because, in this manuscript, some figures (Figure 1-3, 8) were taken by the author, and some figures (Figure 4-7, 9) were already published elsewhere. I believe that a review paper should use summarized tables or figures which were made by author (by that I do not mean pathology or immunohistochemistry data. These are for original research paper). Therefore, I suggest to reorganize the paper.
All of the figures in this paper related to the cornea were from published studies. The only exceptions were the slit lamp photos of corneal fibrosis after alkali burn in Fig. 1E and 1F that were from a preliminary study not being published just to show the effect of another common injury that causes corneal scarring fibrosis. In each of the published studies, each group at each time point would have 4 to 6 corneas analyzed. Therefore, I used other images from these published studies, so I didn’t need to ask permission from the original publication for many of them. I did go back and make sure that the figure legends all include the reference. Some journals, for example Matrix Biology, do not want figures that have been previously published, and that is why I used alternative figures from these studies in this requested Cells submission in many of the cornea figures.
3. Author submitted the manuscript to “cellular molecular mechanisms fibrosis” section. This manuscript is very well-organized, however, the context of manuscript is too focused on a simple explanation. I believe that a review paper should draw new conclusions from the existing data.
Nina Zidar, MD, PhD is editing the special Issue "Cellular and Molecular Mechanisms of Fibrosis" and she asked me to submit a review for this issue related to our work in the cornea that conclusively demonstrates the critical role of the EBM and Descemet’s basement membrane in modulating fibrosis in the cornea for researchers in other fields. It does draw new conclusions related to fibrosis in other organs where the potential role of BM injury and defective regeneration has not been studied or appreciated. I hope this paper will stimulate work that supports or refutes my hypothesis for other organs.
Round 2
Reviewer 2 Report
All questions were addressed.